# Fear Learning Enhances Prefrontal Cortical Suppression of Auditory Thalamic Inputs to the Amygdala in Adults, but Not Adolescents

**DOI:** 10.3390/ijms21083008

**Published:** 2020-04-24

**Authors:** Nicole C. Ferrara, Eliska Mrackova, Maxine K. Loh, Mallika Padival, J. Amiel Rosenkranz

**Affiliations:** 1Department of Foundational Sciences and Humanities, Discipline of Cellular and Molecular Pharmacology, Chicago Medical School, Rosalind Franklin University of Medicine and Science, North Chicago, IL 60064, USA; eliska.mrackova@rosalindfranklin.edu (E.M.); m.loh@my.rfums.org (M.K.L.); mallika.padival@rosalindfranklin.edu (M.P.); jeremy.rosenkranz@rosalindfranklin.edu (J.A.R.); 2Center for Neurobiology of Stress Resilience and Psychiatric Disorders, Rosalind Franklin University of Medicine and Science, North Chicago, IL 60064, USA

**Keywords:** prefrontal cortex, basolateral amygdala, medial geniculate nucleus, adolescent, adult

## Abstract

Adolescence is characterized by increased susceptibility to the development of fear- and anxiety-related disorders. Adolescents also show elevated fear responding and aversive learning that is resistant to behavioral interventions, which may be related to alterations in the circuitry supporting fear learning. These features are linked to ongoing adolescent development of medial prefrontal cortical (PFC) inputs to the basolateral amygdala (BLA) that regulate neural activity and contribute to the refinement of fear responses. Here, we tested the hypothesis that the extent of PFC inhibition of the BLA following fear learning is greater in adults than in adolescents, using anesthetized in vivo recordings to measure local field potentials (LFPs) evoked by stimulation of PFC or auditory thalamic (MgN) inputs to BLA. We found that BLA LFPs evoked by stimulation of MgN inputs were enhanced in adults following fear conditioning. Fear conditioning also led to reduced summation of BLA LFPs evoked in response to PFC train stimulation, and increased the capacity of PFC inhibition of MgN inputs in adults. These data suggest that fear conditioning recruits additional inhibitory capacity by PFC inputs to BLA in adults, but that this capacity is weaker in adolescents. These results provide insight into how the development of PFC inputs may relate to age differences in memory retention and persistence following aversive learning.

## 1. Introduction

Adolescents are particularly vulnerable to stressors that can result in lasting maladaptive behavioral effects associated with increased risk for fear- and anxiety-related disorders [1]. Exposure to stressors during adolescence often results in heightened expression of learned fear and an inability to appropriately suppress fear [2,3,4]. This is frequently reported using aversive learning experiences, where the pairing of a neutral conditional stimulus (CS) with an aversive unconditional stimulus (UCS) forms a memory, and the strength of this memory can be measured with a fear response (e.g., freezing [5]) to the CS in the absence of the UCS during a subsequent test. Using these procedures, adolescents exhibit heightened fear expression during learning as well as at long-term retention tests when compared with adults [6]. The differences in behavioral responding during and following aversive learning between adolescents and adults may be attributed to the regulation of brain regions driving fear responses during adolescence. This highlights a need to understand the brain circuit differences that underlie learning-related changes over the course of development.

Ongoing brain maturation is believed to contribute to behavioral differences between adults and adolescents. The basolateral amygdala (BLA) is essential for fear learning as well as memory formation and maintenance [7,8,9,10,11,12]. Developmental differences in inhibitory processes in the BLA as well as maturation of cortical inputs underlie shifts in fear behavior and responsiveness to stressors [13,14,15,16,17]. The medial prefrontal cortex (PFC) can be divided into the prelimbic (PL) and infralimbic (IL) regions, both of which can modulate BLA neuronal firing [18]. These regions are believed to have opposing roles in behavioral responding, and a balance between activity in these regions shapes the response elicited by a CS [19,20,21,22]. While the PL is believed to be involved in the attention directed at a given stimulus during uncertainty, changes in IL activity direct the context sensitivity of a response, perhaps guided by the strongest CS association (e.g., renewal, and discrimination of over- and under-trained cues) [6,20,23,24]. Many developmental differences are seen between adults and adolescents with regard to IL-dependent behavior [6,25], commonly measured as reduced flexibility in behavioral responding during tasks that require suppression of a learned response, such as fear extinction, and are attributed to the immaturity of IL projections [6,25,26,27]. IL inputs to the BLA are potentiated following initial fear learning and IL—BLA excitability is increased following extinction in adults [28,29], highlighting that IL inputs can change in many conditions where behavioral flexibility is guided by context or competing environmental cues, whether or not suppression of a behavioral response is required [30]. Changes in the IL–BLA pathway may mediate developmental differences in the regulation of fear responses and BLA suppression, where this neuronal inhibition recruited by IL inputs to BLA is essential for reducing conditional fear responses [31,32,33].

While PFC inputs to the BLA are undergoing substantial maturation, learning-related changes in fear responding in adolescents could also be driven by developmental differences in thalamic input and the regulation of these inputs to the BLA. The medial geniculate nucleus (MgN) of the thalamus sends robust projections to the BLA and is critically involved in auditory fear learning and memory [34,35]. Synaptic plasticity and increased strength at MgN–BLA synapses are essential during both learning and memory consolidation, ultimately regulating the degree and precision of fear expression [36,37,38,39]. Further, PFC stimulation inhibits activity evoked in pathways that show learning-related plasticity or in neuronal responses to previously conditioned stimuli in the BLA, suggesting an interaction between the PFC–BLA and MgN–BLA pathways is essential for the generation of learned fear responding that may occur during memory formation [31,33]. Together, this work highlights a fear circuit in adults in which fear learning strengthens the PFC–BLA and MgN–BLA synapses, and the interaction between these inputs on evoked BLA activity may contribute to the degree of fear responding at subsequent memory retention tests. However, it is unclear how these inputs interact in the BLA, particularly in IL–BLA projections in adolescents, to regulate fear responses. Here, we studied how IL regulation of BLA activity changes following fear learning as well as the degree to which these PFC inputs regulate BLA responses to MgN stimulation over the course of development.

## 2. Results

### 2.1. Fear Conditioning Facilitates BLA Responses to MgN Stimulation in Adults, but Not Adolescents

Adult and adolescent rats experienced delay fear conditioning (DFC) or remained naïve. There were no differences in freezing across age in rats that underwent DFC (interaction: F(11, 212.3) = 0.90, *p* = 0.54; main effect of age: F(1, 21) = 1.46, *p* = 0.24; Figure 1B). Naïve and fear conditioned adult and adolescent groups were anesthetized and prepared for in vivo extracellular recordings (Figure 1A). To assess potential developmental alterations in neural circuitry, we recorded BLA local field potentials (LFPs) evoked by a single stimulation in the PFC or the MgN at varying intensities (0.3–1.0 mA) in a counterbalanced manner. The stimulation intensity that evoked an LFP amplitude of approximately half of the maximal amplitude was selected for PFC train stimulation and a subsequent MgN stimulation (Figure 1A).

Changes in evoked BLA LFPs as a result of fear conditioning in adults and adolescents were calculated as a function of MgN or PFC stimulation at varying intensities (single pulse, 0.3–1.0 mA). BLA responses to MgN stimulation (MgN–BLA response) in both adults and adolescents changed as a function of stimulation intensity (adult amplitude: F(1.33, 31.98) = 58.31, *p* < 0.01; adult slope: F(1.24, 27.23) = 4.33, *p* < 0.05, Figure 2B; adolescent amplitude: F(2, 54) = 7.947, *p* < 0.01; adolescent slope: F(1.77, 46.03) = 5.91, *p* < 0.01, Figure 2D). MgN–BLA responses were increased following fear conditioning in adults (main effect of fear conditioning (FC) amplitude: F(1, 24) = 5.26, *p* < 0.05; main effect of FC slope: F(1, 22) = 5.26, *p* < 0.05; Figure 2B), but not adolescents (main effect of FC amplitude F(1, 27) = 0.59, *p* = 0.45; slope: F(1, 26) = 0.03, *p* = 0.86). The main effect of fear conditioning suggests that fear learning facilitates MgN-evoked responses in BLA in adults, but not adolescents. BLA responses to PFC stimulation (PFC–BLA response) in both of adults and adolescents also changed as a function of stimulation intensity (adult amplitude: F(1.611, 38.65) = 51.29, *p* < 0.01; adult slope: F(1.86, 44.70) = 80.14, *p* < 0.0001, Figure 2A; adolescent amplitude: F(1.421, 38.38) = 25.23, *p* < 0.01, adolescent slope: F(2, 50) = 19.51 *p* < 0.001, Figure 2C). Adult and adolescent PFC–BLA responses were not different between FC and naïve groups (main effect: adult slope: F(1, 24) = 0.003, *p* = 0.96; adult amplitude: F(1, 24) = 0.36, *p* = 0.55; adolescent slope: F(1, 25) = 0.91, *p* = 0.35; adolescent amplitude: F(1, 25) = 0.91; *p* = 0.35).

### 2.2. BLA Summation of LFPs Evoked by PFC Train Stimulation in Adults, but Not Adolescents

Alterations in LFP responses over the course of brief train stimulation can indicate recruitment of excitatory and inhibitory components and/or alterations in synaptic strength. Previous work has highlighted developmental differences in PFC–BLA train response [18]. Specifically, developmental differences in the summation of LFPs during 20 Hz train stimulation are evident throughout the BLA, with adolescents showing relatively little summation or suppression from initial stimulation. Because developmental differences are evident throughout the BLA at 20 Hz and 20 Hz is a physiologically relevant cortical firing frequency in awake animals, we used this train frequency to understand how alterations in PFC–BLA synaptic processes change as a result of fear conditioning. We found that 20 Hz train stimulation produced a different response in naïve and fear conditioned groups in adults (main effect of FC: F(1, 21) = 5.1, *p* < 0.05; Figure 3A,D). Adolescents showed relatively little change over the course of the train (Figure 3B).

In order to compare differences in summation or suppression as a result of fear conditioning, the normalized final LFP of each train stimulation was compared between the naïve and fear conditioned groups. To normalize, the final LFP was divided by the first LFP (summation ratio), with numbers greater than 1 indicating facilitation and less than 1 indicating suppression. The summation ratio was impacted by fear conditioning (main effect FC: F(1, 46) = 4.81, *p* < 0.05; interaction: F(1, 46) = 2.89, *p* = 0.096; Figure 3C), but this depended on age such that the summation ratio was higher in naïve adult rats compared with naïve adolescent rats (*p* < 0.05), and fear conditioning decreased the summation ratio in adults (*p* < 0.05), but not adolescents (*p* = 0.73). While this summation ratio can change for a number of reasons, reductions in the adult PFC–BLA summation as a result of fear conditioning may indicate recruitment of inhibitory processes that suppress BLA LFP summation. The absence of summation in adolescents may indicate that PFC inputs to the BLA are weaker due to incomplete maturation, or that competing gamma-Aminobutyric acid (GABA) and glutamate processes recruited by the PFC stimulation balance each other. These processes may contribute to a lack of change during or at the end of the PFC train. To help differentiate between these possibilities, we examined the rapid effect of brief PFC train stimulation on a subsequent LFP.

### 2.3. Developmental Differences in the Suppression of MgN-Evoked BLA LFPs Following Fear Conditioning

Prefrontal cortical inputs regulate BLA activity in a mature circuit, largely through inhibition. During fear conditioning and memory retrieval, BLA activity can be evoked by MgN inputs and activity from PFC inputs may act to suppress MgN-driven BLA activity. Because alterations in the PFC–BLA train response strongly suggest recruitment of inhibition in adults, but not adolescents after fear conditioning, we investigated whether these alterations are sufficient to suppress subsequent MgN–BLA responses. To test this, we added a single MgN stimulation following the PFC train to determine whether the PFC train is sufficient to reduce MgN–BLA responses (example traces Figure 4C).

Both groups increased freezing responses during conditioning (Figure 1B). There were no differences between MgN–BLA responses that followed a low PFC stimulation intensity in adults and adolescents (FC x age: F(1, 46) = 0.007, *p* = 0.934; Figure 4A). However, when PFC stimulation intensity was increased (50% of the maximal response, as seen in Figure 3), there were alterations in the subsequent MgN–BLA response (main effect of age: F(1, 46) = 6.07, *p* < 0.05; FC x age interaction: F(1, 46) = 4.760, *p* < 0.05; Figure 4B). The PFC train resulted in a significant reduction in the MgN–BLA response in adults when compared with adolescents following fear conditioning (*p* < 0.01), and a significant reduction in the MgN–BLA response between naïve and FC adults (*p* < 0.05). Together, this suggests that fear conditioning facilitates PFC regulation of BLA responses to MgN inputs in adults.

### 2.4. Prefrontal Cortical Summation is Associated with BLA Suppression of MgN Input

To account for differences between the ratio of PFC–BLA summation and the degree of the subsequent MgN–BLA response within the same subject, we subtracted the PFC–BLA summation ratio values from the MgN–BLA response. This approach allows us to account for individual differences in the PFC–BLA summation while assessing the PFC effects on the MgN–BLA responses. Here, higher PFC–BLA summation would result in a positive number, whereas lower PFC–BLA summation combined with lower a MgN–BLA response would result in a negative number. Using this approach, we found a significant interaction (F(1, 48) = 4.66, *p* < 0.05, Figure 5A), with lower scores reported in fear conditioned adults when compared with naïve adults (*p* < 0.05). This indicates that the change in summation caused by FC is associated with the effect of PFC stimulation on MgN–BLA responses following fear learning. There was relatively little change in score between adolescent groups.

We next wanted to determine whether MgN–BLA responses following PFC train stimulation were associated with age. To do this, we focused on fear conditioned groups, as the alterations in MgN–BLA responses are more clearly demonstrated in adults following fear learning. We found that the effect of PFC train stimulation on subsequent MgN–BLA response was correlated with development, with fear conditioned adults showing lower MgN–BLA responses than adolescents (F (1, 19) = 6.419, *p* = 0.03, *R*^2^ = 0.25, Figure 5B). Together, this work suggests that a change in PFC–BLA summation may contribute to the degree of the effect of PFC stimulation on the MgN–BLA response, with lower PFC–BLA summation numbers resulting in greater suppression of MgN–BLA responses in adults. This further supports an interpretation that reduced summation of PFC inputs is caused by increased recruitment of inhibitory processes over the course of the PFC train stimulation in adults, while the minimal summation of PFC inputs in adolescents is the result of weaker PFC inputs. Developmental differences in PFC–BLA circuitry likely contribute to changes in the summation ratio, which subsequently influence learning-related changes in MgN–BLA responses, indicated by a correlation between age and MgN–BLA responses following PFC train stimulation (Figure 5C).

## 3. Discussion

Synaptic inputs from auditory thalamic centers to the BLA are strengthened following learning and during memory consolidation. PFC stimulation regulates BLA activity in an age-dependent manner, and this ongoing maturation during adolescence may contribute to the regulation of fear responses after fear learning [6,13,18]. Here, we provide evidence for heightened PFC regulation of the MgN–BLA responses following fear learning in adults, but not adolescents. Specifically, we found that PFC stimulation reduces MgN–BLA responses following fear conditioning in adults. Suppression of MgN–BLA responses following 20 Hz PFC train stimulation was highest in fear conditioned adults. This is consistent with PFC inputs recruiting BLA inhibition that suppresses the summation of LFPs evoked by the PFC or by the MgN. We also found that the reduction in MgN–BLA responses following PFC train stimulation was negatively correlated with age, suggesting that maturation of PFC inputs into the BLA may be a major contributing factor for the regulation of evoked responses in the BLA following fear learning. Collectively, these results suggest that learning-specific PFC regulation of BLA responses to the MgN increase over the course of development (Figure 5C). However, we should note that these differences between fear conditioned and naïve groups in adolescents and adults may in part be due to fear conditioning parameters (e.g., shock intensity or CS–UCS pairings).

Developmental changes in the PFC–BLA pathway are believed to contribute to alterations in the balance of excitation and inhibition in the BLA. Presence of PFC inputs to the BLA are relatively stable from early adolescence to adulthood; however, increases in inhibitory synaptic transmission continue until mid-adolescence (PND 30) [13]. This is supported by prior work demonstrating increases in spontaneous inhibitory post synaptic current frequency until PND 30, unlike spontaneous excitatory post synaptic currents [13]. Changes in the degree of inhibition have also been characterized by the duration of inhibition evoked by PFC stimulation to the BLA, which increases from adolescence to adulthood [18]. Consistent with this, the present results demonstrate a developmental shift in PFC–BLA summation, in addition to the degree to which PFC inputs inhibit MgN–BLA responses following fear learning. Together with prior work, our findings suggest that, as GABAergic processes develop from adolescence into adulthood, the ability of PFC inputs to regulate evoked activity in the BLA following fear learning increases.

In line with the literature highlighting PFC recruitment of inhibition in the BLA, we found that PFC stimulation reduces MgN–BLA responses, an input that is necessary for fear learning and retention. The increase in the ability for the IL to regulate BLA activity following fear learning supports work from others demonstrating increases in IL-evoked field potentials in the BLA following fear learning, as well as the PFC (both PL and IL) suppression of BLA activity in response to previously conditioned stimuli [29,33]. While it is difficult to define the contributions of IL inputs to the BLA following fear conditioning based on these results, the IL has been largely characterized by CS-elicited fear responses that rely on a specific context. With regard to the BLA, IL inputs have been commonly associated with fear response suppression following extinction learning, where a deficiency in the ability to reduce fear responses is seen throughout a majority of the adolescent window [6,25,26]. In adults, there is competition between PL- and IL-inputs that may promote context-specific fear expression and suppression, respectively [20,21,22,24]. However, reduced maturation of PFC inputs to the BLA may drive reduced cognitive flexibility and BLA neuronal suppression in adolescents when fear and extinction memories compete [25]. Our data combined with previous work suggest that fear learning strengthens IL–BLA and MgN–BLA pathways in adults, and IL–BLA pathway strengthening following learning in a mature circuit may promote subsequent inhibition of fear responding (Figure 5c). While we cannot attribute developmental differences in MgN-evoked BLA responses to the PFC pathway alone, the interaction between these pathways following fear learning appears to be associated with reduced PFC–BLA summation in adults only. This suggests that changes in both of these pathways that occur over the course of development may contribute to differences in circuitry underlying fear learning and responding.

Communication between the PFC and BLA occurs, in part, through oscillations, which can be manipulated and evoked using train frequency stimulations to understand the contribution of different oscillatory patterns between the PFC and BLA. Some of these oscillations from the PFC to the BLA are believed to be regulated by and sufficient to recruit inhibitory networks regulating fear expression [40,41]. While the current cortical train stimulation frequency (20 Hz) was chosen based on developmental differences between adults and adolescents seen throughout the BLA, future work should investigate the contribution of various frequency train ranges [18]. It is possible that higher stimulation frequencies or higher stimulation intensities may be able to overcome age differences observed here by recruiting greater inhibitory circuits in the BLA. Synchronization of both theta and gamma oscillations have been closely linked to changes in fear responses to fear and neutral cues [42,43], with theta oscillations closely associated with fear expression and memory formation in the PFC–BLA pathway. Alterations in MgN–BLA responses would likely be influenced by these mechanisms, as activity in these pathways is essential for memory formation and long-term fear responding. Further, higher frequency PFC stimulations (40 Hz) have been shown to evoke substantial inhibition in both adults and adolescents, suggesting that higher frequency stimulations in the PFC–BLA pathway would regulate excitatory evoked activity in the BLA regardless of age, which may contribute to behavioral adaptation [18].

Taken together, we provide support for the interaction between PFC and MgN inputs into the BLA following fear learning. The suppression of MgN–BLA responses was evident following fear conditioning, suggesting the requirement for strengthening of the PFC–BLA pathway underlies optimal fear suppression after fear learning. Additionally, the degree of MgN–BLA suppression increased over the course of development. Together with previous literature, this work suggests the refinement of fear responses from adolescence to adulthood is driven by and associated with maturation of PFC–BLA circuits.

## 4. Materials and Methods

Experiments were approved by the Institutional Animal Care and Use Committee at Rosalind Franklin University (#18-20; August 2, 2018).

### 4.1. Subjects

Subjects were male Sprague Dawley rats purchased from Envigo (*n* = 50; Indianapolis, IN, USA) and housed 2–3 per cage in the Rosalind Franklin University animal facility. Rats had free access to food and water at all times and were maintained on a reverse light cycle (12 h light/dark). Adolescent rats arrived to the animal facility at postnatal day (PND) 20–21, and adults arrived at PND 64–69. Rats acclimated in the colony space for one week prior to recording. At recording, adolescents were between the ages of PND 28 and 40, and adults were between PND 71 and 81.

### 4.2. Fear Conditioning 

Fear conditioning consisted of a six-minute baseline, two conditional stimulus (CS)–unconditional stimulus (UCS) pairings, and a four-minute post period. A 10 second 5 kHz 80 dB pure tone was used as a discrete CS, and a 0.25 mA 0.5 s footshock was used as a UCS (Med Associates Inc, Fairfax, VT). The intertrial interval was 80 s. Freezing was defined as the cessation of movement excluding respiration and was automatically scored through VideoFreeze (Med Associates) calibrated to a trained human observer.

### 4.3. In Vivo Extracellular Surgery and Recording

Rats were anesthetized with urethane (1.5 g/kg in 0.9% saline; intraperitoneal injection). In fear conditioned groups, rats were anesthetized one hour following conditioning. Rats were then mounted onto a stereotaxic device (Kopf Instruments) and body temperatures were maintained at 36–37 °C using a heating pad (Model TC-1000; CWE, Ardmore, PA). Coordinates for surgery were chosen based on the rat brain atlas (Paxinos & Watson, 2007). Adult BLA coordinates were A/P –3.0 mm, M/L +5.0 mm, and D/V –6.5 to 7.2 mm. Adolescent BLA coordinates were A/P –3.0 mm, M/L +4.8 mm, and D/V –6.5 to 7.0 mm. Adult PFC coordinates were A/P +2.7 mm, M/L +0.5 mm, and D/V –5.0 mm; and for adolescents, they were were A/P +2.7 mm, M/L +0.3 mm, and D/V –4.6 mm, targeting the IL region of the PFC. Adult MgN coordinates were A/P –5.3 mm, M/L +2.7 mm, and D/V –7.0 mm; and adolescent MgN coordinates were A/P –5.3 mm, M/L +2.5 mm, and D/V –6.6 mm. Holes were drilled over each location and concentric bipolar stimulation electrodes (MicroProbes, Gaithersburg MD, USA) were slowly lowered to the MgN and PFC and left in place for approximately 45 minutes before recording.

Single-barrel glass recording electrodes were pulled (PE-2; Narishige, Tokyo, Japan) and broken under a microscope for a 1–2 μm diameter tip. The electrode was then filled with Pontamine (2% Chicago Sky Blue 6B, Sigma-Aldrich, St Louis MO, USA) in 2 M NaCl. The electrode was mounted onto the stereotaxic device and slowly lowered into the brain, targeting the BLA, with a hydraulic microdrive (SKU 50-12-9-02 and SKU 50-12-1C; Frederick Haer & Co, Bowdoin, ME, USA). Signals were amplified via a headstage preamplifier and filtered via an amplifier (Model 1800; A-M Systems, Sequim WA) at a low cut-off frequency of 0.1 Hz and a high cut-off frequency of 5 kHz. Signals were transmitted to an audio monitor (Model AM7, Grass Medical Instruments, Warwick, RI, USA). Following filtering from the amplifier, digitized outputs were recorded and monitored on a personal computer (Mac Pro; Apple, Cupertino, CA, USA) using Axograph X software (Sydney, Australia) and stored for later analysis.

### 4.4. Local Field Potential (LFP) Recordings

The MgN and PFC were stimulated via concentric bipolar electrodes controlled by a pulse stimulator (Grass S88, Grass Instruments, Warwick, RI, USA). LFPs as a result of single stimulation (0.2 Hz, 0.3–1.0 mA, 0.2 ms duration) of the PFC or MgN were recorded in the BLA, and a stimulation intensity of approximately half of the amplitude measured at 1.0 mA was used for subsequent recording protocols. When specified, train stimulations of the PFC occurred at 20 Hz (10 pulses/train, 50 ms inter-stimulus interval, 0.1–0.7 mA). To understand the degree of PFC influence on BLA activity, the stimulation intensity of 0.1 mA for the PFC train was also used to assess minimal BLA responses to PFC stimulation. MgN single stimulation occurred 100 ms following PFC train. The duration of the stimulation protocol was five seconds. Each stimulation protocol was run a minimum of 20 sweeps (trials). Each stimulation intensity was then averaged and analyzed. The order of stimulation protocol (0 mA PFC train, 0.1 mA PFC train, 50% PFC train) was counterbalanced between rats in all groups.

### 4.5. Histology 

At the conclusion of the recording experiments, Pontamine dye was ejected from the recording electrode (–30 μA, 30 minutes) via constant current source. Brains were removed and fixed in 4% paraformaldehyde (Electron Microscopy Sciences, Hatfield, PA, USA) overnight. Brains were transferred to 0.1 M phosphate buffered saline until sections were collected (60 μm) on a vibratome (VT1000 S, Leica, Wetzlar, Germany). Sections containing PFC, BLA, and MgN placements were mounted onto subbed slides and stained with cresyl violet. Stimulation and recording placements were reconstructed based on the histological staining and confirmed by borders defined in a rat brain atlas (Paxinos & Watson, 2007).

### 4.6. Data Analysis

BLA responses to MgN stimulation are referred to as MgN–BLA responses, while PFC single or train stimulation evoked BLA LFPs are referred to as PFC–BLA responses or PFC–BLA summation, respectively. Completed sweeps of stimulation protocols were averaged and filtered (20 sweeps, 0.1 kHz). Evoked LFP slopes were measured via Axograph X software (Sydney, Australia). For input/output curves, slopes and amplitudes were normalized to 1.0 mA to calculate a 50% response from maximal stimulation intensity (amplitude_XmA_/amplitude_1.0mA_). For train stimulation, slopes were normalized to the first evoked LFP (slope_X_/slope_1_). The final BLA slope ratio in the PFC train was graphed and averaged to measure summation (>1) or suppression (<1) [18]. MgN–BLA responses (slope) after the PFC train stimulation were normalized to the MgN–BLA responses in the absence of PFC stimulation. To further understand how PFC train stimulation influences BLA responses to MgN stimulation, a change score was created accounting for the change in MgN–BLA response and PFC–BLA summation. These values were calculated as the difference between the normalized MgN–BLA response (in the presence or absence of PFC train) and subtracted from the PFC–BLA summation ratio (last minus first response in the train, Figure 4A).

### 4.7. Statistical Analysis 

Statistical analyses were performed in IBM SPSS, and graphs were made using Prism 8 software (GraphPad, La Jolla, CA, USA). Statistical significance was defined as a *p*-value less than 0.05. A two-way analysis of variance (ANOVA) with age (adult, adolescent) or stimulation intensity and behavioral manipulation (naïve, fear conditioned) as main factors was used when stated. A repeated measures ANOVA was used when stimulation intensity or stimulation number in the train was a main factor. Pairwise comparisons were used to follow up on the main effects or interactions. Corrected Levene’s values were used when variances were not equal. Simple linear regression was used to determine the relationships between age and BLA slope in response to MgN stimulation, as well as BLA summation during PFC train and BLA slope in response to MgN stimulation. All graphed data are presented as mean with SEM. Outliers were removed if they were more than two standard deviations from the mean.

## Figures and Tables

**Figure 1 ijms-21-03008-f001:**
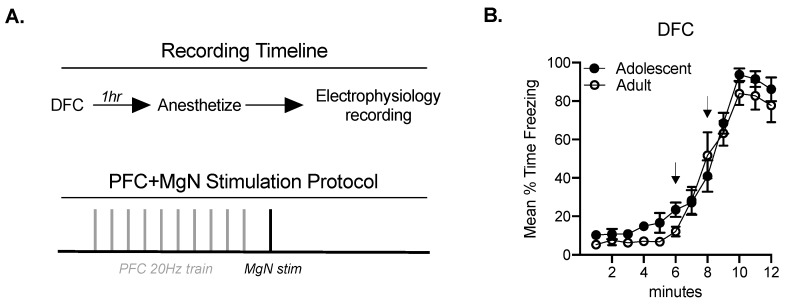
Delay fear conditioned or naïve groups were anesthetized and prepared for in vivo recordings, where a 20 Hz prefrontal cortex (PFC) train stimulation preceded a single MgN stimulation (**A**). Delay fear conditioning (DFC) data is graphed in 60 s bins and for both adults (*n* = 12) and adolescents (*n* = 15). This consisted of a 6 min baseline, two conditional stimulus (CS)–unconditional stimulus (UCS), and a 4 min post. Arrows indicate the period in which a CS-UCS pairing occurred. The mean percent time freezing increased during and following CS–UCS pairings (**B**).

**Figure 2 ijms-21-03008-f002:**
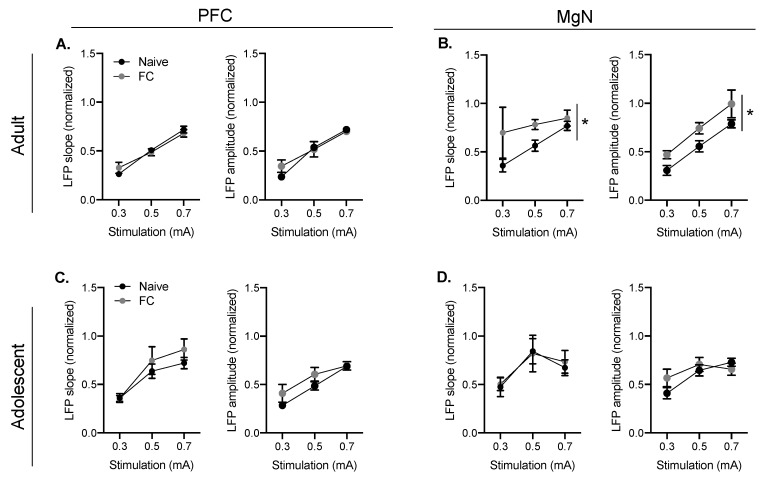
Fear conditioning (FC) facilitates MgN–basolateral amygdala (BLA) responses in adults, but not adolescents. Local field potentials (LFPs) in the BLA following MgN or PFC stimulation (0.3–1.0 mA) of adult and adolescent rats in naïve and fear conditioned groups were collected. The amplitude and slope of PFC–BLA increased with increasing PFC stimulation intensities in both of adults (naïve n = 16, FC *n* = 10) and adolescents (naïve *n* = 16; FC *n* = 13) (**A**,**C**). MgN–BLA responses increased with increasing MgN stimulation intensities in adults, and these responses were higher following FC in adults (naïve *n* = 16, FC *n* = 10) (**B**). MgN–BLA responses increased with increasing stimulation intensities in adolescents; however, there were no differences between naïve and FC groups (naïve n = 15; FC *n* = 14) (**D**). * indicates significant main effect between naïve and fear conditioned groups when *p* < 0.05.

**Figure 3 ijms-21-03008-f003:**
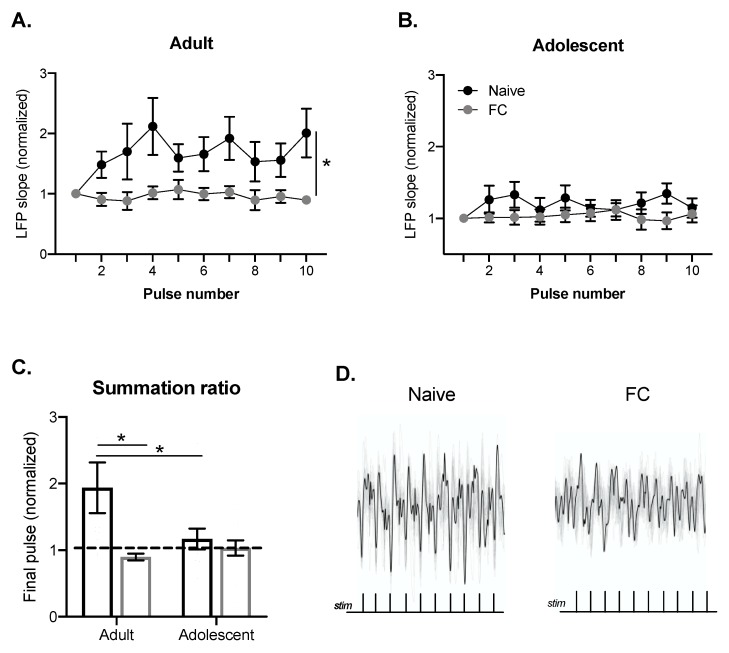
BLA summation of LFPs evoked by PFC train stimulation in adults, but not adolescents. Stimulation intensities that evoked half of the maximal PFC–BLA response (based on responses at 1.0 mA) were used for PFC stimulation. The slope for each PFC–BLA response was calculated after each stimulation in the 20 Hz train. All slopes were normalized to the first PFC–BLA response of the train stimulation. Naïve adults show increases in PFC–BLA responses over the course of the train, indicating summation that is reduced following FC (**A**). Adolescents show relatively little change in PFC–BLA responses over the course of the train in both naïve and FC groups (**B**). There was a significant difference in the summation ratio (final PFC–BLA slope in the 20 Hz train) of the naïve and FC in adults (naïve n = 15, FC *n* = 9), but not adolescents (naïve n = 11; FC *n* = 13), with FC adults showing reduced summation (**C**). Representative traces of adult PFC–BLA responses during the PFC train (**D**). * *p* < 0.05.

**Figure 4 ijms-21-03008-f004:**
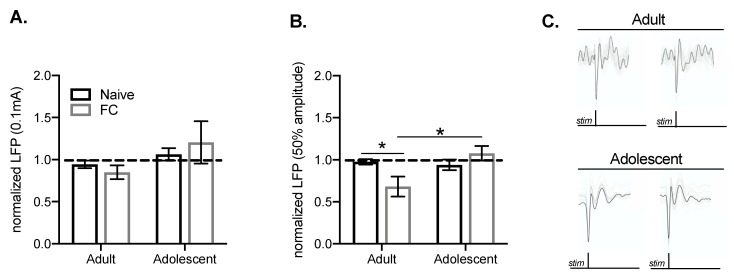
Developmental differences in the suppression of MgN-evoked LFPs in the BLA following fear conditioning. MgN–BLA responses following PFC train stimulation were measured to assess the influence of PFC stimulation on evoked BLA responses. There were no differences in MgN–BLA responses following PFC stimulation at a low intensity (**A**). There was a significant reduction in the MgN–BLA response following higher PFC train stimulation (50% maximal amplitude of 1.0 mA) in adults after FC (*n* = 8) when compared with naïve (*n* = 16) adult groups, as well as between FC adult and adolescent groups (*n* = 13; adolescent naïve *n* = 12) (**B**). Representative traces of MgN–BLA LFPs (**C**). * *p* < 0.05.

**Figure 5 ijms-21-03008-f005:**
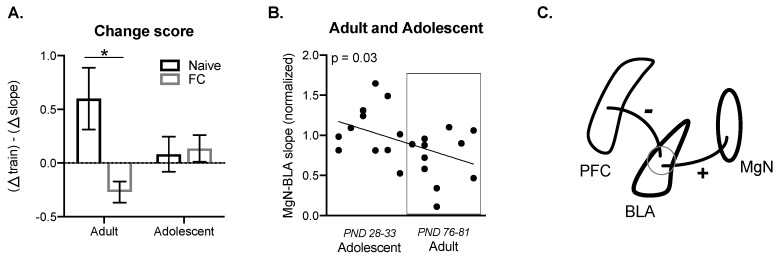
Prefrontal cortical summation is associated with BLA suppression of MgN input. A change score was created to measure within subject differences in PFC–BLA responses to train stimulation and subsequent MgN–BLA responses. There was a significant difference between FC (*n* = 9) and naïve adults (*n* = 15), but not adolescents (naïve n = 13; FC *n* = 12) (**A**). Following FC, the effect of PFC train stimulation on a subsequent MgN–BLA response was correlated with age, with adults showing lower MgN–BLA responses when compared with adolescents (adult *n* = 11; adolescent *n* = 12) (**B**). Schematic of hypothesized PFC–MgN–BLA interactions during development following fear learning, where PFC–BLA responses are measured as the summation ratio and the change in summation ratio is related to the degree of suppression of MgN–BLA responses. Suppression of MgN responses by PFC inputs may be related to maturation of PFC inputs to the BLA through recruitment of inhibitory processes (**C**). * *p* < 0.05.

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
