# Peer review of "Fear Learning Enhances Prefrontal Cortical Suppression of Auditory Thalamic Inputs to the Amygdala in Adults, but Not Adolescents"

_ijms, 2020, doi:10.3390/ijms21083008_

Round 1

Reviewer 1 Report

The manuscript by Ferrara and colleagues examines developmental changes in the circuitry underlying the encoding of auditory fear conditioning in rats. They assessed changes in prefrontal cortex (PFC)- and medial geniculate nucleus (MgN) - basolateral amygdala (BLA) responses using local field potentials (LFPs) following auditory fear conditioning in adolescent (PND28-40) and adult (PND71-81) rats.  The authors first explored changes in evoked BLA local field potentials (LFPs) following MgN or PFC stimulation after fear learning in adults versus adolescents. One hour following training, MgN-BLA LFP responses were potentiated in fear conditioned adults but not adolescents. They found no change in PFC-BLA responses. Next, they compared summation or suppression of BLA LFPs following fear conditioning in adults versus adolescents. They saw that fear conditioning blunted summation driven by 20 Hz train stimulation in BLA in adults but not adolescents. This is likely driven by the fact that adolescents show very low summation in naïve conditions, as the group had previously shown (Seleck et al., 2018). In addition, PFC train resulted in a reduction in subsequent MgN-BLA response in fear conditioned adults but not adolescents, suggesting adolescents lack fear-induced facilitation of PFC regulation of MgN-BLA responses.

Research regarding PFC-BLA circuit maturation is still growing, and the use of extracellular electrophysiology provides an important tool to investigate differences across development. Overall, the manuscript shows for the first time that while adults demonstrate enhanced PFC regulation of MgN-BLA activity after fear conditioning, adolescent rats lack auditory fear-induced regulation of PFC mediated modulation of MgN-BLA responses, which could have implications for developmental differences in fear encoding and expression.

Major concerns:

  • My main concern is that while the authors base their rationale for looking at fear-related developmental changes around extinction, which differs between adolescents and adults, they examine changes in PFC-BLA inputs 1h after learning, a time in which the behavioral output of fear learning is not changed between these ages. Adolescent rats have been shown to display resistance to extinction (although the authors do not provide behavioral data here), however the authors do not explore extinction-induced changes. Similarly, it is unclear whether the authors are examining IL- or PL-BLA responses, which have opposite roles in acquisition/extinction (see comment below). The manuscript would be strengthened by a more targeted rationale and hypothesis surrounding their results. For instance, is their main claim that fear encoding differs in adolescence at the circuit level, and that is what leads to extinction deficits following acquisition? If so, has this PFC-mediated facilitation of Mgn-BLA transmission ever been shown to be functionally relevant for fear expression/extinction success? How would this feature into our understanding of fear encoding and extinction? Although their finding is very interesting, and adds to a literature that needs precisely this type of experiments, clarification on the implications of their findings as well as their general hypotheses would significantly strengthen the manuscript.

  • As mentioned, the main focus of this manuscript is on the inhibitory actions of the PFC and its connections to the BLA; however, the authors did not specify which region of the PFC they targeted during their experiments. Given that PL-BLA projections have been implicated in fear expression, while IL-BLA projections are involved in suppression (or extinction) of the fear memory, stating which subregion is being targeted is necessary to clarify their results. The coordinates listed in the methods section suggest they are targeting IL, but this needs to be explicitly stated.

  • There is a mismatch between the authors’s claim on the contribution of inhibition to fear-induced plasticity in PFC-BLA synapses and what has been reported, at least in the slice electrophysiology literature. This needs to be addressed in several points of the manuscript. For example, the authors state in the introduction that “This is further supported by extensive evidence demonstrating the necessity for local plasticity in the PFC as well as at PFC-BLA synapses during extinction learning through the recruitment of inhibitory processes (Amano et al., 2010; Bloodgood et al., 2018; Sierra-Mercado et al., 2011; Vieira et al., 2015). In this case, activity at PFC-BLA synapses are believed to suppress activity through excitation of local GABAergic interneurons, highlighting that dynamic changes in local BLA activity can be regulated by PFC inputs (Cho et al., 2013; Rosenkranz & Grace, 2002; Rosenkranz et al., 2003).”. Cho and colleagues show in their study that there are no changes in inhibition following extinction learning in PFC-BLA synapses (the shift in inhibition:excitation is driven by changes in excitation), nor extinction-driven changes in PFC-ITC synapses that could affect BLA (Cho et al., 2013). As such, I suggest the authors re-work these sentences to reflect this, as there is less direct support for a role of inhibition in this process as is suggested here. They repeat something similar in the discussion: “These local signals are believed to be transmitted to the BLA to aid in the refinement of fear responses to danger or safe cues largely through the recruitment of inhibition, which has also been demonstrated during extinction learning.”. The authors need to either provide further justification for this claim, or modify it to be more consistent with the cited literature.

  • Similarly, the authors say that “PFC inputs to the BLA are believed to contribute to the refinement of fear responses following learning, which can be through suppression of BLA activity (Bloodgood et al., 2018; Rosenkranz et al., 2003; Vieira et al., 2015)”. Bloodgood et al. do not show evidence of suppression of BLA activity following learning. They record from IL or PL neurons projecting to BLA and therefore offer no direct evidence to that effect. Vieira et al., also show no direct evidence to support this statement. These references should be corrected or the statement nuanced.

Minor points (suggestions)

  • Sample size data is missing for individual experiments. Plotting individual data points would also increase transparency in the data.

  • Addition of an unpaired control, as well as behavioral data from a parallel cohort of animals showing fear retrieval and extinction in adolescents and adults would strengthen the manuscript.

Author Response

We sincerely thank the reviewer for the thoughtful comments. We have now edited the manuscript to address these critiques. In summary, we have substantially edited the manuscript to correct the inappropriate focus on extinction, as raised by both reviewers, added clarifications, corrected citations, and rearranged data presentation. We believe that these reviewer comments have led to improvements in our manuscript. A point-by-point response to each critique is below.

Major concerns:

  • My main concern is that while the authors base their rationale for looking at fear-related developmental changes around extinction, which differs between adolescents and adults, they examine changes in PFC-BLA inputs 1h after learning, a time in which the behavioral output of fear learning is not changed between these ages. Adolescent rats have been shown to display resistance to extinction (although the authors do not provide behavioral data here), however the authors do not explore extinction-induced changes. Similarly, it is unclear whether the authors are examining IL- or PL-BLA responses, which have opposite roles in acquisition/extinction (see comment below). The manuscript would be strengthened by a more targeted rationale and hypothesis surrounding their results. For instance, is their main claim that fear encoding differs in adolescence at the circuit level, and that is what leads to extinction deficits following acquisition? If so, has this PFC-mediated facilitation of Mgn-BLA transmission ever been shown to be functionally relevant for fear expression/extinction success? How would this feature into our understanding of fear encoding and extinction? Although their finding is very interesting, and adds to a literature that needs precisely this type of experiments, clarification on the implications of their findings as well as their general hypotheses would significantly strengthen the manuscript.

Response: We have made changes throughout the manuscript to reframe our findings and minimize implications for extinction and correct miscited references. Instead, we now focus on changes in MgN-BLA and PFC-BLA responses following learning with implications for memory formation. The one-hour time point was chosen based on finding that AMPA:NMDA ratios are increased in the BLA following FC as early as 5m following conditioning and persist for several days (e.g. Hong et al., 2013) and that this time point has been used to characterize the boundary of memory consolidation (e.g. Bourtchouladze et al., 1998) while still observing peaks in plasticity markers (e.g. Jarome et al., 2011). We have added points to highlight alterations in this circuitry following learning and during a consolidation window that may inform developmental differences in other behaviors.

We made edits throughout the document addressing concerns from both reviewers about the overly heavy focus on extinction. This has been tapered or removed, and we have refocused these sections of the paper to reframe our findings highlighting the developmental differences in circuitry following fear learning.    

  • Abstract lines 14-15 and 26-27.
  • Page 1 lines 37-40.
  • Page 2 lines 8, 9, 12, 14, 19-23.
  • Page 7 lines 1-3, 28, 34.
  • Page 8 lines 3-4.

  • As mentioned, the main focus of this manuscript is on the inhibitory actions of the PFC and its connections to the BLA; however, the authors did not specify which region of the PFC they targeted during their experiments. Given that PL-BLA projections have been implicated in fear expression, while IL-BLA projections are involved in suppression (or extinction) of the fear memory, stating which subregion is being targeted is necessary to clarify their results. The coordinates listed in the methods section suggest they are targeting IL, but this needs to be explicitly stated.

Response: We thank the reviewer for pointing out this missing information. We used coordinates from prior publications in the lab targeted the IL region of the PFC (Selleck et al., 2018) and have added this information to the manuscript on page 8 (lines 33-34).

  • There is a mismatch between the authors’s claim on the contribution of inhibition to fear-induced plasticity in PFC-BLA synapses and what has been reported, at least in the slice electrophysiology literature. This needs to be addressed in several points of the manuscript. For example, the authors state in the introduction that “This is further supported by extensive evidence demonstrating the necessity for local plasticity in the PFC as well as at PFC-BLA synapses during extinction learning through the recruitment of inhibitory processes (Amano et al., 2010; Bloodgood et al., 2018; Sierra-Mercado et al., 2011; Vieira et al., 2015). In this case, activity at PFC-BLA synapses are believed to suppress activity through excitation of local GABAergic interneurons, highlighting that dynamic changes in local BLA activity can be regulated by PFC inputs (Cho et al., 2013; Rosenkranz & Grace, 2002; Rosenkranz et al., 2003).”. Cho and colleagues show in their study that there are no changes in inhibition following extinction learning in PFC-BLA synapses (the shift in inhibition:excitation is driven by changes in excitation), nor extinction-driven changes in PFC-ITC synapses that could affect BLA (Cho et al., 2013). As such, I suggest the authors re-work these sentences to reflect this, as there is less direct support for a role of inhibition in this process as is suggested here. They repeat something similar in the discussion: “These local signals are believed to be transmitted to the BLA to aid in the refinement of fear responses to danger or safe cues largely through the recruitment of inhibition, which has also been demonstrated during extinction learning.”. The authors need to either provide further justification for this claim, or modify it to be more consistent with the cited literature.

  • Similarly, the authors say that “PFC inputs to the BLA are believed to contribute to the refinement of fear responses following learning, which can be through suppression of BLA activity (Bloodgood et al., 2018; Rosenkranz et al., 2003; Vieira et al., 2015)”. Bloodgood et al. do not show evidence of suppression of BLA activity following learning. They record from IL or PL neurons projecting to BLA and therefore offer no direct evidence to that effect. Vieira et al., also show no direct evidence to support this statement. These references should be corrected or the statement nuanced.

Response: We thank the reviewer for these comments, and we have edited several sections in the manuscript to incorporate these comments, and have corrected citations (page 2 lines 12-24).

Minor points:

  • Sample size data is missing for individual experiments. Plotting individual data points would also increase transparency in the data.

Response: Sample sizes have been added to figure captions.

  • Addition of an unpaired control, as well as behavioral data from a parallel cohort of animals showing fear retrieval and extinction in adolescents and adults would strengthen the manuscript.

Response: While this raises an interesting point and would highlight developmental differences in learned behavioral responses, we have edited the manuscript to focus on developmental differences in the circuitry following fear learning, during a memory consolidation window. We appreciate the suggestion regarding an unpaired control group. This type of control was not added in this instance based on prior work that unpaired controls do not result in synaptic plasticity at thalamo-BLA synapses. Based on this, we think it is unlikely that the alterations in PFC-BLA and MgN-BLA responses are due to tone or shock exposure alone, and instead are likely based on the pairings of these events.

References

Bourtchouladze R, Abel T, Berman N, Gordon R, Lapidus K, & Kandel ER. (1998). Different training procedures recruit either one or two critical periods for contextual memory consolidation, each of which requires protein synthesis and PKA. Learning & Memory, 5: 365-374.

Hong I, Kim J, Kim J, Lee S, Ko H, Nader K, Kaang B, Tsien RW, & Choi S. (2013). AMPA receptor exchange underlies transient memory destabilization on retrieval. PNAS 110 (20): 8218-8223.

Jarome TJ, Werner CT, Kwapis JL, & Helmstetter FJ. (2011). Activity dependent protein degradation is critical for the formation and stability of fear memory in the amygdala. PLoS ONE, 6(9): e24349.

Selleck RA, Zhang W, Samberg HD, Padival M, & Rosenkranz JA. (2018). Limited prefrontal cortical regulation over the basolateral amygdala in adolescent rats. Scientific reports.

Reviewer 2 Report

The manuscript by Ferrara etal offers a new relationship between delay fear conditioning behavior and suppression of auditory thalamic inputs to the amygdala in male rats, with distinct responses in adults vs. adolescent subjects.

The manuscript is mostly well written and addresses an important question. Some issues need to be addressed.

A figure with behavioral data needs to be presented prior to any electrophysiological data. For example, freezing responses are only shown in figure 3. This information needs to be moved to a figure indicating behavior and time line of experimental manipulation. In addition, it will be important to show freezing behavior –per minute data- during six minute baseline, first and second CS-UCS responses and the last 4 minute post period. Since the overall behavior did not differ between the age groups, it will be interesting to see if there were any age differences during the 4 minute post period when compared by a minute data. Also, it will help to have a schematic for the time line for the electrophysiological recordings in naïve and FC animals.  

The y-axis for figure 2 should be the same for the adult and adolescent groups. In addition, it is not clear if there are differences in LFP slope between adolescents and adults in the naïve condition and whether this difference contributed to the lack of effect in FC animals (a threshold effect). This needs to be discussed.

In the discussion section, some sentences could be re-written for improved clarity.

E.g. lines 30 to 31, Page 6, Suppression of MgN-BLA responses by ….. were highest in adults with less PFC-BLA summation, the meaning of this sentence is not clear.

E.g. Lines 1-2 page 7, differences may in part….. sensitivity to the chosen shock intensity… This issue is not clear. If there were differences in sensitivity would it not be evident in freezing behavior?

Also, there is quite a bit of discussion on the role of PFC inputs to the BLA and the role in extinction. However, no extinction was performed in adolescent and adult rats, therefore, this discussion should be tapered.

Author Response

We sincerely thank the reviewer for the thoughtful comments. We have now edited the manuscript to address these critiques. In summary, we have substantially edited the manuscript to correct the inappropriate focus on extinction, as raised by both reviewers, added clarifications, corrected citations, and rearranged data presentation. We believe that these reviewer comments have led to improvements in our manuscript. A point-by-point response to each critique is below.

  • A figure with behavioral data needs to be presented prior to any electrophysiological data. For example, freezing responses are only shown in figure 3. This information needs to be moved to a figure indicating behavior and time line of experimental manipulation. In addition, it will be important to show freezing behavior –per minute data- during six minute baseline, first and second CS-UCS responses and the last 4 minute post period. Since the overall behavior did not differ between the age groups, it will be interesting to see if there were any age differences during the 4 minute post period when compared by a minute data. Also, it will help to have a schematic for the time line for the electrophysiological recordings in naïve and FC animals.  

Response: We thank the reviewer for thoughtful critiques and comments. We have rearranged data presented in the figures to present behavioral data first. This training data is graphed in minute-by-minute points. While adolescents appear to freeze slightly more, this is not statistically significant from adults. This new figure also includes a schematic of behavior and recording procedures. A paragraph discussing this graph has been added to page 2 (lines 37-39) and 3 (lines 1-6). As a result of shifting this to the first figure, previous figure numbers have been shifted.

  • The y-axis for figure 2 should be the same for the adult and adolescent groups. In addition, it is not clear if there are differences in LFP slope between adolescents and adults in the naïve condition and whether this difference contributed to the lack of effect in FC animals (a threshold effect). This needs to be discussed.

Response: We have corrected to y axis and added a sentence (p5 line 1-4, now labeled Figure 3) describing differences in the LFP slope between age groups in the naïve condition. This is also reflected in Figure 3. With regards to threshold effect, we think the reviewer is suggesting that differences in the impact of PFC inputs across age might be due to weaker PFC effects even in the naïve adolescents – but if we stimulated at higher intensities we may be able to overcome this deficiency in adolescents. This is a possibility. However, we do not believe that this alone would account for differences in the impact of PFC inputs. Thus, the single pulse stimulation results suggest similar LFPs can be evoked by PFC stimulation in adults and adolescents, and we chose train stimulation intensities based on this single pulse data. Despite this similarity, there were still age differences in the effects of PFC trains. However, we fully agree that the age differences might be able to be overcome if enough BLA inhibitory processes are recruited (either higher stimulation intensity or a higher stimulation frequency). We have now acknowledged this important issue in the Discussion (pg 7, lines 42-44)

  • In the discussion section, some sentences could be re-written for improved clarity.
    • g. lines 30 to 31, Page 6, Suppression of MgN-BLA responses by ….. were highest in adults with less PFC-BLA summation, the meaning of this sentence is not clear.
    • g. Lines 1-2 page 7, differences may in part….. sensitivity to the chosen shock intensity… This issue is not clear. If there were differences in sensitivity would it not be evident in freezing behavior?

Response: This has been corrected, and the edited sentence is on pg 6 (lines 36-37).  We agree with the reviewer that it is likely that differences in foot shock sensitivity would emerge as differences in freezing during fear conditioning. Our intention here is to be conservative, and recognize that the behavioral procedure itself might exert age-dependent effects on the neural circuitry, even it does not produce differences in our measures of freezing during conditioning.  Shock intensity and number of CS-UCS pairings are just meant as examples of key parameters in that can dictate degree of freezing. This sentence on p7 (line 4-7) has been edited to reflect this.

  • Also, there is quite a bit of discussion on the role of PFC inputs to the BLA and the role in extinction. However, no extinction was performed in adolescent and adult rats, therefore, this discussion should be tapered.

Response: We thank the reviewer for this comment, and we made edits throughout the document addressing concerns from both reviewers about the heavy focus on extinction. This has been tapered or removed. We have refocused these sections of the paper to reframe our findings highlighting the developmental differences in circuitry following fear learning.    

  • Abstract lines 14-15 and 26-27.
  • Page 1 lines 37-40.
  • Page 2 lines 8, 9, 12, 14, 19-23.
  • Page 7 lines 1-3, 28, 34.
  • Page 8 lines 3-4.

Round 2

Reviewer 1 Report

The revised version of the manuscript by Ferrara and colleagues has addressed some of my points by removing some text and reframing their rationale towards fear encoding over extinction. Their main finding is interesting and opens doors to novel investigations of pathways through which developmental changes in fear learning might occur. However, they still have not sufficiently developed their rationale as noted in my review, particularly as it relates to the prefrontal cortex subregions targeted, as well as the relationship between their findings and behavior.

Major concerns:

The authors should explicitly discuss and formulate a hypothesis as to how IL (and not PL, which is more implicated in fear encoding) modulation of the MgN-BLA pathway during encoding might affect extinction. Given that (1) a role for PL-BLA, but not IL-BLA pathway has been shown for fear encoding, (2) that adolescent rats do not show differences in fear acquisition or retrieval compared to adult rats, and that (3) the authors do not offer or cite functional evidence implicating the [IL-(MgN-BLA)] pathway in fear encoding or extinction, the manuscript would benefit from further strengthening the rationale behind the impact and implications of their findings.   

The authors should explicitly say that they are targeting IL in the body of the manuscript and especially in the discussion (not just in the methods), as this targeting has different implications for our understanding of this circuit. This selectivity in targeting is a strength of the manuscript, and should be highlighted, as very little is known about fear learning-induced plasticity in the IL-BLA pathway per se.

Minor concerns:

The authors introduce the following new sentence: “Together, this work highlights a fear circuit in adults in which fear learning strengthens MgN-BLA synapses, which occurs during memory consolidation, while PFC inputs contribute to the inhibition and flexibility of fear responses.”. There is overwhelming evidence of prefrontal cortex being necessary for the encoding of fear memories. This should be modified.

The new sentence “Further, priming of PFC inputs in the BLA is sufficient to reduce responses evoked by auditory cortical inputs in the BLA, suggesting an interaction between PFC and primary auditory pathways may occur during memory formation (Cho et al., 2013).” is not very convincing, as the experiment by Cho et al., found heterosynaptic plasticity following fear learning stimulation of external capsule, whereas MgN projections go through internal capsule.

Author Response

We thank the reviewer for the thoughtful comments. We hope we have addressed the below concerns. Our edits can be seen in several sections in the introduction and discussion section that are highlighted in yellow.

The revised version of the manuscript by Ferrara and colleagues has addressed some of my points by removing some text and reframing their rationale towards fear encoding over extinction. Their main finding is interesting and opens doors to novel investigations of pathways through which developmental changes in fear learning might occur. However, they still have not sufficiently developed their rationale as noted in my review, particularly as it relates to the prefrontal cortex subregions targeted, as well as the relationship between their findings and behavior.

Major concerns:

The authors should explicitly discuss and formulate a hypothesis as to how IL (and not PL, which is more implicated in fear encoding) modulation of the MgN-BLA pathway during encoding might affect extinction. Given that (1) a role for PL-BLA, but not IL-BLA pathway has been shown for fear encoding, (2) that adolescent rats do not show differences in fear acquisition or retrieval compared to adult rats, and that (3) the authors do not offer or cite functional evidence implicating the [IL-(MgN-BLA)] pathway in fear encoding or extinction, the manuscript would benefit from further strengthening the rationale behind the impact and implications of their findings.   

Response: We thank the reviewer for these points. We have added several sentences in the introduction elaborating on IL and PL differences and a rationale for targeting the IL (page 2 lines 5-24). This is largely based on work demonstrated changes in IL activity that determine contextual sensitivity of behavioral responses. Developmental differences emerge with these IL-dependent tasks and can be attributed to reduced flexibility of responding. One of the ways this can be studied following learning is through extinction where responses adapt to changes in the environment. While we did not find developmental differences during fear conditioning, our fear conditioning protocol was not designed to necessarily capture age differences, but instead to understand how the mPFC interacts with pathways essential for eliciting CS-responses after CS-UCS associations are learned in both adults and adolescents.

The authors should explicitly say that they are targeting IL in the body of the manuscript and especially in the discussion (not just in the methods), as this targeting has different implications for our understanding of this circuit. This selectivity in targeting is a strength of the manuscript, and should be highlighted, as very little is known about fear learning-induced plasticity in the IL-BLA pathway per se.

Response: We have included this in the introduction on page 2 line 36, and we have added a section to the discussion section explicitly stating we have targeted the IL and discussed the implications for these findings on page 8 lines 19-35. Some of the points discussed include the role of the IL in the generation of responses that are dependent on the context. This has been commonly studied through extinction where responding is highly dependent on the context in which extinction occurred and rebound of fear can occur in novel or in the original trained context (e.g. renewal). During adolescence differences in fear expression have been attributed to competing memories that are formed during adolescence, potentially driven by IL maturation.

Minor concerns:

The authors introduce the following new sentence: “Together, this work highlights a fear circuit in adults in which fear learning strengthens MgN-BLA synapses, which occurs during memory consolidation, while PFC inputs contribute to the inhibition and flexibility of fear responses.”. There is overwhelming evidence of prefrontal cortex being necessary for the encoding of fear memories. This should be modified.

Response: We have edited this sentence on page 2 lines 38-40 to state that the interaction between PFC and MgN inputs to the BLA may contribute to the degree of fear based on PFC regulation of neuronal responses to conditioned stimuli as well as PFC regulation of pathways that show learning-related changes.

The new sentence “Further, priming of PFC inputs in the BLA is sufficient to reduce responses evoked by auditory cortical inputs in the BLA, suggesting an interaction between PFC and primary auditory pathways may occur during memory formation (Cho et al., 2013).” is not very convincing, as the experiment by Cho et al., found heterosynaptic plasticity following fear learning stimulation of external capsule, whereas MgN projections go through internal capsule.

Response: We have edited this sentence to highlight that the PFC may regulate BLA CS-responsive neurons, and may therefore regulate MgN-BLA pathway activity following learning, as this pathway has been highly implicated in auditory fear learning (page 2 lines 34-38).